# Uncertainty-aware Parameter-Efficient Self-training for Semi-supervised Language Understanding

**Jianing Wang**$^{\diamond}$  **Qiushi Sun**$^{\diamond\heartsuit}$  **Nuo Chen**$^{\diamond}$

**Chengyu Wang**$^{\clubsuit}$ **Jun Huang**$^{\clubsuit}$ **Ming Gao**$^{\diamond\heartsuit*}$ **Xiang Li**$^{\diamond}$

$^{\diamond}$School of Data Science and Engineering, East China Normal University

$^{\heartsuit}$National University of Singapore $^{\clubsuit}$Alibaba Group

$^{\heartsuit}$KLATASDS-MOE, School of Statistics, East China Normal University

lygwjn@gmail.com, {qiushisun,nuochen}@stu.ecnu.edu.cn

{chengyu.wcy,junhuang.hj}@alibaba-inc.com, mgao@dase.ecnu.edu.cn

## Abstract

The recent success of large pre-trained language models (PLMs) heavily hinges on massive labeled data, which typically produces inferior performance in low-resource scenarios. To remedy this dilemma, we study self-training as one of the predominant semi-supervised learning (SSL) approaches, which utilizes large-scale unlabeled data to generate synthetic examples. However, too many noisy labels will hurt the model performance, and the self-training procedure requires multiple training iterations making it more expensive if all the model parameters of the PLM are updated. This paper presents UPET, a novel **U**ncertainty-aware **P**arameter-**E**fficient self-**T**raining framework to effectively and efficiently address the labeled data scarcity issue. Specifically, we incorporate Monte Carlo (MC) dropout in Bayesian neural network (BNN) to perform uncertainty estimation for the teacher model and then judiciously select reliable pseudo-labeled examples based on confidence and certainty. During the student training, we introduce multiple parameter-efficient learning (PEL) paradigms that allow the optimization of only a small percentage of parameters. We also propose a novel Easy-Hard Contrastive Tuning to enhance the robustness and generalization. Extensive experiments over multiple downstream tasks demonstrate that UPET achieves a substantial improvement in terms of performance and efficiency. Our codes and data are released at https://github.com/wjn1996/UPET.

## 1 Introduction

Pre-trained language models (PLMs) have become the imperative infrastructure in a series of downstream natural language understanding (NLU) tasks (Devlin et al., 2019; Liu et al., 2019; Yang et al., 2019), aiming at capturing prior knowledge by pre-training over large-scale unsupervised corpora and fine-tuning on the target tasks. However, the conventional fine-tuning approaches heavily depend on the time-consuming and labor-intensive process of data annotation, which could be even more bothersome in some real-world scenarios and typically produces inferior performance in few-shot settings (Liu et al., 2021b; Kojima et al., 2022).

Recently, self-training (Chawla and Karakoulas, 2005; Amini et al., 2022) has been presented to address the labeled data scarcity issue by leveraging the large-scale unlabeled data in addition to labeled data, which is one of the mature paradigms in semi-supervised learning (Qi and Luo, 2022; Yang et al., 2021a; Chawla and Karakoulas, 2005; van Engelen and Hoos, 2020; Yang et al., 2021b). A *teacher* model is fine-tuned on the few-shot labeled data, then the pseudo label of each unlabeled example can be generated. After that, a *student* model can learn the knowledge derived from the large-scale pseudo-labeled data, leading to better performance near to full-supervised learning. Previous works typically use self-training in conjunction with large PLMs to endow the model with the ability of few-shot learning. Despite the big success, we observe that there are still two challenges. 1) The pseudo-labeled data consists of too many noises, inevitably degrading the model performance due to confirmation bias (Wang et al., 2021). 2) The procedure of self-training is too expensive when updating all parameters of the large PLM [1] (Wang et al., 2022).

Fortunately, parameter-efficient learning (PEL) opens up the possibility of attaining near state-of-the-art performance, whilst adding only a few parameters per task (Mao et al., 2022; Ding et al., 2023; Zhang et al., 2023). Notable PEL-based methods include Ptuning (Liu et al., 2021b), Prefix-

---

* * Corresponding Author.

[1]Generally, the number of training pseudo-labeled data for the student model is larger than labeled data.

tuning (Li and Liang, 2021), Adapter (Houlsby et al., 2019), BitFit (Zaken et al., 2022), LoRA (Hu et al., 2022), etc. Yet, it is unclear how these PEL-based methods can be applied to self-training.

In this paper, we develop a novel **U**ncertainty-aware **P**arameter-**E**fficient self-**T**raining framework (UPET) for improving self-training through two perspectives, i.e., effectiveness and efficiency. To reach these goals, we respectively present two novel techniques, including *Reliable Example Sampling* (RES) and *Efficient Robust Tuning* (ERT). The goal of RES is to explicitly mitigate the effect of label noises. Concretely, we obtain the prediction probability distribution over all unlabeled data derived from the teacher model. Then, we utilize Monte Carlo (MC) dropout technique in Bayesian neural network (BNN) (Gal and Ghahramani, 2016; Wang and Yeung, 2016) to estimate the uncertainty of each unlabeled example. To this end, the example with higher confidence and certainty will be judiciously selected as the reliable pseudo-labeled data. In ERT, we aim to leverage PEL paradigms to train a robust student model over reliable pseudo-labeled data. We design multiple PEL-based model architectures for the student model that only need to update a small scope of tunable parameters in PLM during iterative self-training. Additionally, we introduce Easy-Hard Contrastive Tuning to improve the robustness of the parameter-efficient model, which can be viewed as a regularization in the semantic space that keeps the noisy labels away from the reliable examples.

We conduct extensive experiments over multiple NLU tasks. Results show that UPET outperforms strong baselines in terms of both effectiveness and efficiency. The improvement is consistent in different settings with different PEL methods and the number of labeled data. Our key contributions to this field are summarized as follows: 1) We use parameter-efficient learning of PLMs in conjunction with uncertainty estimation to form an efficient and effective self-training framework. 2) To better improve the robustness of the parameter-efficient model, we introduce Easy-Hard Contrastive Learning. 3) Extensive experiments among a wide range of tasks demonstrate that our proposed framework outperforms prevailing strong baselines.

## 2 Related Work

**Semi-supervised Learning and Self-training.** SSL aims to effectively utilize unlabeled data in addition to labeled data, which has been widely used in the NLP community (Yang et al., 2017; Gururangan et al., 2019; Xie et al., 2020; Chen et al., 2020). For instance, Yang et al. (2017); Gururangan et al. (2019) utilize variational autoencoders (VAEs) for sequence classification and labeling. Chen et al. (2020) proposes MixText to mix labeled, unlabeled, and augmented data, and performs similar consistency training as UDA (Xie et al., 2020). Self-training is one of the mature SSL approaches that use *teacher-student* architecture to augment data (Hu and Khan, 2021; Mukherjee and Awadallah, 2020; Amini et al., 2022; Wang et al., 2021; Tsai et al., 2022). For example, Hu and Khan (2021) presents uncertainty estimation for denoising self-training. Tsai et al. (2022) introduces graph-based contrastive learning to preserve consistency regularization. Wang et al. (2021) incorporates self-training into sequence labeling tasks by automatic weighting strategy.

**Parameter-Efficient Learning.** PEL is to optimize a small portion of parameters while keeping the model backbone frozen, which aims at improving the training efficiency and preserving the model's effectiveness (He et al., 2022). Houlsby et al. (2019) integrates task-specific neural modules called *adapters* into PLMs, and only these *adapters* are updated during fine-tuning. Ptuning (Liu et al., 2021b) and Prefix-Tuning (Li and Liang, 2021) respectively introduce a lightweight prefix module into the input layer and each transformer layer, enabling efficient training over these prefix modules. Notable PEL-based models also include BitFit (Zaken et al., 2022), LoRA, etc. This paper integrates PEL into self-training to improve its efficiency.

## 3 UPET: The Proposed Method

Given a labeled set $\mathcal{D}_l = \{(X_i, Y_i)\}_{i=1}^{N_l}$ and an unlabeled set $\mathcal{D}_u = \{\widetilde{X}_i\}_{i=1}^{N_u}$, where $N_l$ and $N_u$ respectively denote the number of labeled set and unlabeled set ($N_l \ll N_u$). $X_i, \widetilde{X}_i \in \mathcal{X}$ denote the input sentence in the labeled set and unlabeled set, respectively. $Y_i \in \mathcal{Y}$ is the corresponding label of $X_i$. The task is to train a neural model $f^W$ and pseudo label for each unlabeled example $\widetilde{X}_i$, where $f^W : \mathcal{X} \rightarrow \mathcal{Y}$ is a function with parameters $W$ to map the input space $\mathcal{X}$ to the label space $\mathcal{Y}$. We aim to answer the following research problem:

- **RQ1**: How can we mitigate the problem of noisy pseudo labels via judiciously selecting

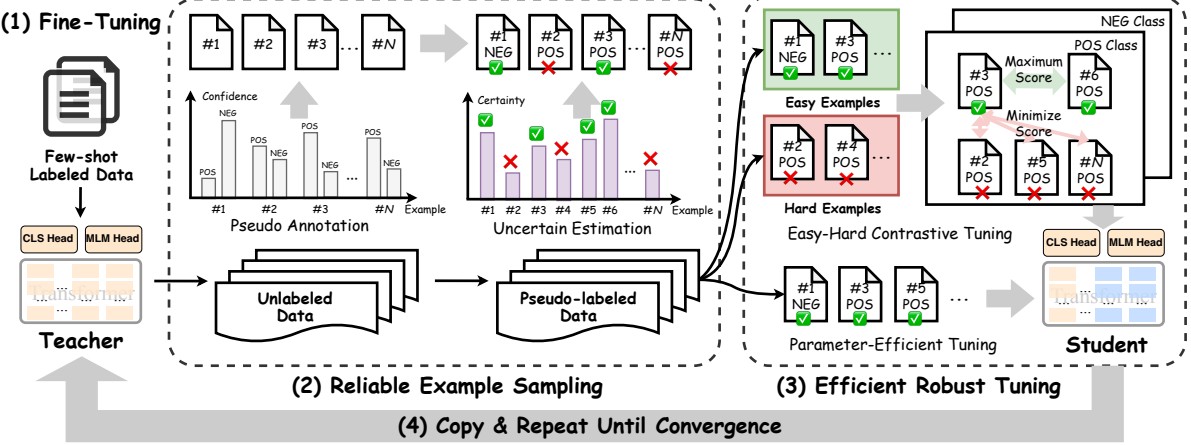

Figure 1: The overview of UPET framework. We first fine-tune a teacher model over few-shot labeled data. Then, we aim to judiciously choose suitable pseudo-labeled data by uncertainty estimation. During student learning, we leverage the parameter-efficient method with robust PHCE loss and contrastive regularization to train the student model on pseudo-labeled data. At last, the student model can be used for the next iteration. (Best viewed in color.)

reliable examples?

- **RQ2**: How can the model parameters be efficiently updated during the iterative self-training process, meanwhile preserving the model's robustness and performance?

We thus propose the UPET framework which consists of two novel techniques, i.e., *Reliable Example Sampling* (RES) and *Efficient Robust Tuning* (ERT). The framework overview is illustrated in Figure 1 and the detailed algorithm is shown in Appendix B.

### 3.1 Fine-Tuning and Pseudo Annotation

We start with a fine-tuning stage over the few-shot labeled data $\mathcal{D}_l$ to form a *teacher* model $f_{tea}^W$. After that, the pseudo label $\widetilde{Y}_i$ of each unlabeled example $\widetilde{X}_i$ can be generated by the teacher model:

$$\widetilde{Y}_i = \arg\max_c p(y = c | f_{tea}^W(\widetilde{X}_i)), \qquad (1)$$

where $p(\cdot)$ is the probability distribution. However, the generated labels may be wrong due to the model confirmation bias problem. That means we need to explicitly reduce the noise problem by designing a suitable sample selection strategy.

### 3.2 Reliable Example Sampling

To reach this goal, we follow Tsai et al. (2022); Mukherjee and Awadallah (2020); Hu and Khan (2021) to leverage uncertainty estimation from BNN to measure what the *reliable* unlabeled examples can be selected for training. we follow (Houlsby et al., 2011; Gal et al., 2017; Tsai

et al., 2022) to leverage *information gain* of the model parameters to show how certain the model is to the pseudo-labeled examples w.r.t. the true labels [2]. Typically, the information gain can be defined as:

$$\mathbb{B}(\widetilde{Y}_i, W | \widetilde{X}_i, \mathcal{D}_u) = \mathbb{H}(\widetilde{Y}_i | \widetilde{X}_i, \mathcal{D}_u) - \mathbb{E}_{p(W|\mathcal{D}_u)}[\mathbb{H}(\widetilde{Y}_i | \widetilde{X}_i, W)], \qquad (2)$$

where $W$ denotes the parameters of the teacher. $\mathbb{B}(\widetilde{Y}_i, W | \widetilde{X}_i, \mathcal{D}_u)$ denotes the information gain which is the difference between $\mathbb{H}(\widetilde{Y}_i | \widetilde{X}_i, \mathcal{D}_u)$ (the final entropy after seeing all examples from unlabeled sentences) and $\mathbb{H}(\widetilde{Y}_i | \widetilde{X}_i, W)$ (the current entropy for the example $\widetilde{X}_i$). $p(W|\mathcal{D}_u)$ is the posterior distribution. As the calculation of Eq. 2 is intractable, we utilize MC dropout in BNN to perform approximation. Specifically, we assume that the posterior distribution $p(W|\mathcal{D}_u)$ can be replaced with dropout distribution $q_\theta(W)$. Thus, we can sample $T$ masked model weight $\{\widetilde{W}_t\}_{t=1}^T \sim q_\theta(W)$, and calculate the approximation value as:

$$\hat{\mathbb{B}}(\widetilde{Y}_i, W | \widetilde{X}_i, \mathcal{D}_u) = -\sum_{c \in \mathcal{Y}} (\frac{1}{T}\sum_{t=1}^T \hat{p}_c^t) \log(\frac{1}{T}\sum_{t=1}^T \hat{p}_c^t) + \frac{1}{T}\sum_{t=1}^T \sum_{c \in \mathcal{Y}} \hat{p}_c^t \log(\hat{p}_c^t), \qquad (3)$$

where $\hat{p}_c^t = p(y_i = c | f_{tea}^{\widetilde{W}_t}(\widetilde{X}_i))$ is the predict probability of $\widetilde{X}_i$ derived from the $t$-th masked model

---

[2]The model certainty can be used to estimate the reliability of the unlabeled example, even though the label is unknown.

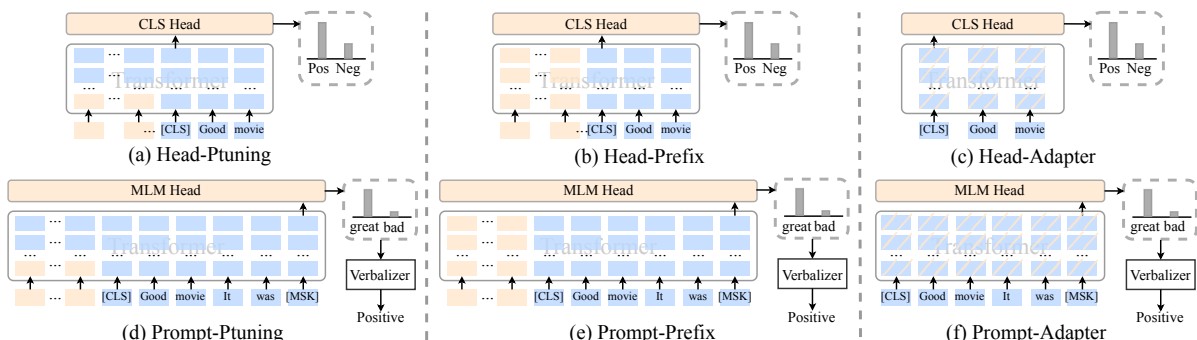

Figure 2: Overview of different PEL paradigms. (a)-(c) represent **Head-Tuning**, aiming to CLS head for prediction. (d)-(f) denote **Prompt-Tuning** to make prediction via well-designed template and verbalizer. We unify three classic PEL methods for both Head-Tuning and Prompt-Tuning. The block in light yellow and blue means the trainable and frozen parameters, respectively. The block with sketches denotes the adapter module. (Best viewed in color.)

$f_{tea}^{\widetilde{W_t}}$. Thus, a lower $\hat{\mathbb{B}}(\tilde{Y}_i, W|\widetilde{X}_i, \mathcal{D}_u)$ value means that the model is more certain about the prediction, as higher certainty corresponds to lower information gain (Tsai et al., 2022) [3]. Formally, we can design a certainty score for each example as:

$$s_i^{ct} = 1 - \hat{\mathbb{B}}(\widetilde{Y}_i, W|\widetilde{X}_i, \mathcal{D}_u). \qquad (4)$$

To this end, we can obtain the final sampling weight for each example by considering both model confidence and certainty:

$$s_i = \frac{\alpha \times s_i^{cf} + (1-\alpha) \times s_i^{ct}}{\sum_{\widetilde{X}_i \in \mathcal{D}_u} \alpha \times s_i^{cf} + (1-\alpha) \times s_i^{ct}}, \qquad (5)$$

where $s_i^{cf} = \frac{1}{T}\sum_{t=1}^{T} p(y = \widetilde{Y}_i|f_{tea}^{\widetilde{W_t}}(\widetilde{X}_i))$ is the model confidence derived from the average approximate posterior of the $T$ masked models w.r.t the pseudo label $\widetilde{Y}_i$, $\alpha$ ($0 \leq \alpha \leq 1$) denotes the balancing factor. Hence, a number of $N_r$ reliable examples can be sampled by these weights to form a new subset $\mathcal{D}_r \subset \mathcal{D}_u$.

### 3.3 Efficient Robust Tuning

#### 3.3.1 Parameter-Efficient Tuning

After the annotation and selection of unlabeled examples, we need to train a student model to elicit knowledge from the teacher. Yet, the training process of the self-training paradigm is inefficient. To remedy this dilemma, we aim to introduce PEL in self-training. We initialize a student model $f_{stu}^{W^*}$ and a few designated parameters in $W^*$ can be tuned, enabling efficiency when training on many

pseudo-labeled data. To meet our desiderata, we introduce two prediction paradigms with three PEL methods. The architecture is shown in Figure 2.

**Head-Tuning.** Head-Tuning leverages CLS head to generate the probability distribution of the given example. Formally, we have:

$$p_{W^*}(y|\widetilde{X}_i) = \mathcal{H}_{cls}(\mathcal{F}_{W^*}(\widetilde{X}_i)), \qquad (6)$$

where $\mathcal{F}_{W^*}(\cdot)$ denotes the output representation by the student model $f_{stu}^{W^*}$. $\mathcal{H}_{cls}(\cdot)$ denotes a CLS head with a softmax classification layer [4].

**Prompt-Tuning.** Prompt-Tuning aims at reusing the Masked Language Modeling (MLM) head to make predictions. Specifically, a well-designed template $\mathcal{T}$ with a masked token ("[MASK]") is concatenated with the original input sentence. In addition, we need to define a verbalizer $\mathcal{V}$ that maps the probability distribution over the whole vocabulary set $\mathcal{X}$ to the label set $\mathcal{Y}$. The probability can be calculated as:

$$p_{W^*}(y|\widetilde{X}_i) = \mathcal{V}_y(\mathcal{H}_{mlm}(\mathcal{F}_{W^*}(\mathcal{T}||\widetilde{X}_i))), \quad (7)$$

where $\mathcal{H}_{mlm}$ denotes the MLM head derived from the PLM, $\cdot||\cdot$ is the concatenation operation. $\mathcal{V}_y(\cdot)$ aims to map the label word's probability at the masked position to the corresponding class $y$.

Hence, we can integrate Ptuning (Liu et al., 2021b), Prefix-tuning (Li and Liang, 2021) and Adapter-tuning (Houlsby et al., 2019) to unify the PEL with arbitrary PLMs and prediction paradigms, including Head-Ptuning, Head-Prefix, Head-Adapter, Prompt-Ptuning, Prompt-Prefix and

---

[3] Intuitively, if the model is always certain about some examples, these examples might be too easy to contribute any additional information.

[4] It can be viewed as a feed-forward network (FFN) with a random initialized parameters.

Prompt-Adapter. More details are shown in Appendix A.1. During the optimization, we can compute the following cross-entropy objective by:

$$l(\mathcal{D}_r, f_{stu}^{W^*}) = \frac{1}{N_r} \sum_{(\widetilde{X}_i, \widetilde{Y}_i) \in \mathcal{D}_r} \log p_{W^*}(y = \widetilde{Y}_i | \widetilde{X}_i). \quad (8)$$

Yet, it is still possible that the subset $\mathcal{D}_r$ could consist of some wrong labels. During the parameter-efficient training stage, the scale of trainable parameters in $W^*$ being small, the student model is fragile and the robustness could not be preserved due to the negative effect of these noises in the backward. In that, we follow (Tsai et al., 2022) to utilize partially huberised cross-entropy loss (PHCE loss), which is an alternative variant with a gradient clipping technique. Hence, the loss function in Eq. 8 can be modified as:

$$l(\mathcal{D}_r, f_{stu}^{W^*}) = \frac{1}{N_r} \sum_{(\widetilde{X}_i, \widetilde{Y}_i) \in \mathcal{D}_r} \phi_\tau(y = \widetilde{Y}_i | \widetilde{X}_i), \quad (9)$$

where $\phi_\tau(y|x)$ is the PHCE loss function with a hyper-parameter $\tau$ ($\tau > 1$). The detail of the PHCE loss function is shown in Appendix A.3.

### 3.3.2 Easy-Hard Contrastive Tuning

As mentioned above, the selected example in $D_r$ has a higher model certainty and might be too *easy* to contribute any additional information. Nonetheless, this inevitably leads to the student model *over-fitting* on these frequently selected samples (Mukherjee and Awadallah, 2020). Intuitively, the example not selected in $\mathcal{D}_r$ is more likely to be a noise that results in semantic drift. Thus, a natural idea is to exploit some *hard* examples (which are not selected in $\mathcal{D}_r$) as the negatives to keep them away from *easy* (reliable) examples, which can be viewed as a regularization in the semantic space.

To reach this goal, we present Easy-Hard Contrastive Tuning. We denote $\mathcal{D}_h$ as the difference between $\mathcal{D}_u$ and $\mathcal{D}_r$, so the examples in $\mathcal{D}_h$ represent the *hard* ones. During the optimization of the student model, given one example $(\widetilde{X}_i, \widetilde{Y}_i) \in \mathcal{D}_r$, we aim to choose one another example $(\widetilde{X}_i^+, \widetilde{Y}_i^+)$ from $\mathcal{D}_r$ as the positive and some negative examples $\{(\widetilde{X}_{ik}^-, \widetilde{Y}_{ik}^-)\}_{k=1}^{N_n}$ from $\mathcal{D}_h$, where $N_n$ is the number of negatives, $\widetilde{Y}_i = \widetilde{Y}_i^+ = \widetilde{Y}_{ik}^-$ have the same class [5]. Hence, the contrastive regularization

---

[5] The pseudo label of the hard example may be wrong, so if the sampled hard example has the same label with $(\widetilde{X}_i, \widetilde{Y}_i)$, it can be viewed as a negative in terms of the class $\widetilde{Y}_i$.

term can be computed as:

$$R(f_{stu}^{W^*}) = \frac{1}{N_r} \sum_{c \in \mathcal{Y}} \sum_{(\widetilde{X}_i, \widetilde{Y}_i) \in \mathcal{D}_r, \widetilde{Y}_i = c}$$
$$\left[ \frac{\exp(g(\widetilde{X}_i, \widetilde{X}_i^+))}{\exp(g(\widetilde{X}_i, \widetilde{X}_i^+)) + \frac{1}{N_n} \sum_{k=1}^{N_n} \exp(g(\widetilde{X}_i, \widetilde{X}_{ik}^-))} \right], \quad (10)$$

where $g(\cdot, \cdot)$ is the score function that measures the similarity of two examples in the semantic space. Finally, the whole training objective is designed as:

$$\mathcal{L}(\mathcal{D}_r, f_{stu}^{W^*}) = l(\mathcal{D}_r, f_{stu}^{W^*}) + \lambda R(f_{stu}^{W^*}), \quad (11)$$

where $\lambda > 0$ is the hyper-parameter.

## 4 Experiments

### 4.1 Dataset and Implementation Details

We perform extensive experiments over seven language understanding tasks to evaluate our UPET framework. We choose a series of tasks from the GLUE benchmark (Wang et al., 2018), including SST-2 (Socher et al., 2013) for sentiment analysis, MNLI (Williams et al., 2018) for language inference, QNLI (Rajpurkar et al., 2016) for question answering, MRPC (Dolan and Brockett, 2005) for semantic paraphrasing and RTE (Dagan et al., 2005) for textual entailment. We also choose CB (De Marneffe et al., 2019) from SuperGLUE (Wang et al., 2019) for linguistic entailment and AGNews (Zhang et al., 2015) for topic classification. For each dataset, the number of labeled examples per class is set as $N_l \in \{16, 32, 64\}$. We repeatedly sample few-shot labeled instances five times with different seeds from $\{12, 21, 42, 87, 100\}$ and report average performance with standard deviation.

For the implementation details, we choose RoBERTa-large (Liu et al., 2019) from Hugging-Face [6] as the default backbone for both the teacher and student model. The number of the self-training iterations is set as 5. We train models by the AdamW algorithm with $\beta_1 = 0.9, \beta_2 = 0.98$ on 4 NVIDIA V100-32G GPUs. For each task, we use grid search to select the best hyper-parameter (Appendix D). By default, the training epoch of the teacher and student are 100.

---

[6] https://huggingface.co/transformers/index.html.

| Baselines | Use PEL | #Tunable Params. | SST-2 (acc) | MNLI (acc) | QNLI (f1) | MRPC (acc) | RTE (acc) | CB (acc) | AGNews (acc) | Avg. |
|---|---|---|---|---|---|---|---|---|---|---|
| *Full Data* | | | | | | | | | | |
| Head FT | ✗ | 355M | 95.2 | 89.8 | 93.3 | 91.4 | 83.0 | 90.5 | 94.7 | 91.1 |
| Prompt FT | ✗ | 355M | 95.9 | 90.2 | 93.0 | 90.9 | 88.4 | 91.1 | 94.0 | 91.9 |
| *Few Labeled Data (16-shot)* | | | | | | | | | | |
| Head FT | ✗ | 355M | 81.4±3.8 | 45.8±6.4 | 60.2±6.5 | 75.9±2.9 | 54.4±3.9 | 74.5±2.6 | 88.9±2.7 | 68.7 |
| Prompt FT | ✗ | 355M | 90.6±1.1 | 53.7±2.3 | 64.5±4.0 | 74.4±3.0 | 59.1±3.6 | 77.0±3.3 | 88.6±1.2 | 72.6 |
| *Few Labeled Data (16-shot) + Unlabeled Data* | | | | | | | | | | |
| Head ST | ✗ | 355M | 87.9±3.0 | 51.9±2.8 | 64.0±2.8 | 79.4±2.5 | 53.2±2.9 | 75.9±1.5 | 86.4±3.0 | 71.2 |
| Prompt ST | ✗ | 355M | 91.0±3.1 | 57.7±2.9 | 67.8±3.2 | 81.0±2.4 | 57.9±3.3 | 77.7±2.9 | 88.8±3.5 | 74.6 |
| UST | ✗ | 355M | 84.0±4.0 | 53.9±2.9 | 65.9±3.3 | 79.9±2.0 | 55.6±2.6 | 76.0±3.1 | 89.3±3.5 | 72.1 |
| CEST | ✗ | 355M | 86.4±3.8 | 52.2±2.9 | 65.0±2.4 | 80.8±3.5 | 57.0±1.9 | 78.1±2.7 | 88.5±2.2 | 72.6 |
| LiST | ✓ | 14M | 91.0±3.0 | 62.0±3.9 | 67.4±2.5 | 82.0±3.3 | 60.8±2.5 | 79.7±2.9 | 90.3±2.5 | 76.2 |
| UPET | | | | | | | | | | |
| - Head-Ptuning | ✓ | <1M | 90.8±3.2 | 53.2±2.9 | 64.8±2.8 | 82.6±2.8 | 59.3±3.7 | 76.8±2.6 | 90.8±1.8 | 74.0 |
| - Head-Prefix | ✓ | <6M | 87.5±2.0 | 56.7±2.7 | 69.2±3.1 | 82.3±2.2 | 58.7±2.5 | 79.6±1.5 | 90.9±1.8 | 74.6 |
| - Head-Adapter | ✓ | 14M | 89.3±1.0 | 60.1±2.6 | 68.5±1.4 | **85.5±2.5** | 59.2±3.5 | 79.0±1.5 | 90.3±2.6 | 76.0 |
| - Prompt-Ptuning | ✓ | <1M | 91.7±2.8 | **69.5±1.9** | **71.9±2.8** | 83.7±3.3 | 60.8±1.5 | 80.4±1.4 | 89.6±2.2 | **78.2** |
| - Prompt-Prefix | ✓ | <6M | **92.3±2.0** | 64.2±2.9 | 66.1±3.0 | 83.0±1.8 | **61.5±1.6** | **80.8±2.1** | 90.5±3.1 | 76.9 |
| - Prompt-Adapter | ✓ | 14M | 91.9±1.9 | 66.1±2.9 | 66.8±1.8 | 84.2±1.4 | 61.0±1.6 | 80.4±2.0 | **91.0±2.0** | 77.3 |

Table 1: The performance comparison of accuracy or F1 scores (%) with standard deviations on seven tasks. All methods (except fine-tuning with full data) are trained with 16-shot labeled samples for each class and overall results are aggregated over five different runs with different random seeds. In UPET, the first three variants belong to the Head-Tuning paradigm, while the others are Prompt-Tuning.

## 4.2 Baselines

We consider some strong baselines for comparison, including **UST** (Mukherjee and Awadallah, 2020), **CEST** (Tsai et al., 2022) and **LiST** (Wang et al., 2022). UST and CEST leverage uncertainty estimation for self-training. LiST integrates Adapter-tuning (Houlsby et al., 2019) into prompt-based learning for parameter-efficient self-training, which is similar to the Prompt-Adapter paradigm. In addition, we also design two semi-supervised learning baselines: 1) **Head ST** aims to use the classic fine-tuning with CLS head to augment unlabeled data through standard self-training. 2) **Prompt ST** aims to reuse the MLM head with a well-designed task-specific template and verbalizer to perform pseudo-labeling in standard self-training. We also choose **Head FT** and **Prompt FT** to fine-tune over few-shot or full training data.

## 4.3 Main Results

Table 1 illustrates the main results over seven NLU tasks with different settings. RoBERTa-large trained on fully labeled examples provides the ceiling performance for the few-shot and semi-supervised setting. We thus make the following observations. 1) According to the overall results, all the methods with self-training outperform conven-

tional few-shot learning (i.e., Head FT and Prompt FT). This demonstrates the impact of self-training with unlabeled data. 2) We obtain the best overall performance of 78.2% with the lowest tunable parameters (i.e., Prompt-Ptuning) and improve over Head ST, Prompt ST, UST, CEST, and LiST by 7.0%, 3.6%, 6.1%, 5.6%, and 2.0% respectively over seven tasks, which indicates that UPET outperforms state-of-the-arts in terms of both the effectiveness and efficiency. 3) Compared to the strong baseline Prompt ST that uses the PEL-based approach, we obtain a 3.6% absolute improvement, demonstrating the substantial contributions of the well-designed reliable example selection and contrastive regularization. 4) We also list all 6 PEL paradigms' performance of UPET. We observe that the performance of Prompt-Tuning is higher than Head-Tuning, indicating that reusing the pre-training objective MLM with the task-orient template and verbalizer is more effective for self-training. In addition, more tunable parameters may enhance the student model's ability to learn semantic knowledge derived from the teacher.

## 4.4 Further Analysis

**Impact of Self-training Iterations.** To validate the effectiveness of self-training, we choose MNLI

| Teacher Use PEL | Student Use PEL | # Tunable Params. | Avg. Result | Avg. Time |
|---|---|---|---|---|
| *Head-Adapter* | | | | |
| ✗ | ✗ | 355M+355M | 76.6 | 11.3h |
| ✗ | ✓ | 355M+14M | 76.0 | 4.1h |
| ✓ | ✗ | 14M+355M | 75.2 | 10.7h |
| ✓ | ✓ | 14M+14M | 75.0 | 3.8h |
| *Prompt-Adapter* | | | | |
| ✗ | ✗ | 355M+355M | 77.6 | 11.0h |
| ✗ | ✓ | 355M+14M | 77.2 | 4.0h |
| ✓ | ✗ | 14M+355M | 76.4 | 10.7h |
| ✓ | ✓ | 14M+14M | 75.8 | 3.9h |

Table 2: The average performance (%) over all tasks with different combinations of PEL paradigms.

and RTE and draw some curves to show the performance of different PEL paradigms at each iteration in Figure 3.

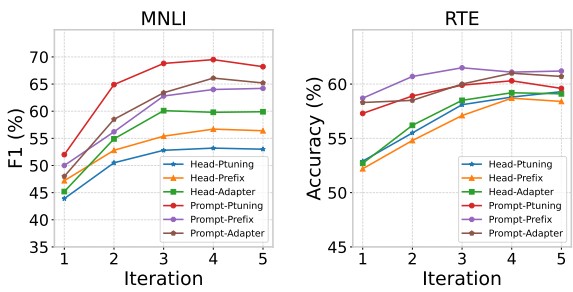

Figure 3: The performance (%) of different self-training iterations over MNLI and RTE.

From the figure, we find that the performance increases when the framework continual training until the 4-th iteration, indicating the convergence of our framework. Additionally, the student model with Prompt-Tuning (including Prompt-Ptuning, Prompt-Prefix, and Prompt-Adapter) consistently outperforms Head-Tuning (including Head-Tuning, Head-Prefix, and Head-Adapter). This shows that prompt-based methods can better utilize PEL to make self-training both effective and efficient.

**Labeled Data Efficiency.** To investigate the influence of the number of labeled examples, we vary the examples of each class from 16, 32, and 64.

We choose LiST as the strong baseline. To make a fair comparison, the PEL we select is Prompt-Adapter, which is the same as LiST and only tunes the adapter module in PLM. Results in Table 3 illustrate that the performance gradually improves as the number of labeled data increases, as expected. In addition, we also find that our UPET outper-

| #-shot⟶ | LiST | | | UPET | | |
|---|---|---|---|---|---|---|
| | 16 | 32 | 64 | 16 | 32 | 64 |
| SST-2 | 91.0 | 91.8 | 92.7 | **91.9** | **93.0** | **93.6** |
| MNLI | 62.0 | 65.7 | 69.7 | **66.1** | **69.2** | **72.3** |
| QNLI | **67.4** | **71.5** | 74.4 | 66.8 | 71.1 | **75.0** |
| MRPC | 82.0 | 84.2 | **85.8** | **84.2** | **85.1** | 85.7 |
| RTE | 60.8 | 64.2 | 67.9 | **61.0** | **66.0** | **68.9** |
| CB | 79.7 | 83.1 | 85.7 | **80.4** | **84.3** | **86.2** |
| AGNews | 90.3 | 90.8 | 91.3 | **91.0** | **91.4** | **91.9** |

Table 3: The performance (%) with different numbers (16/32/64 examples per class) of labeled data. The parameter-efficient paradigm is Prompt-Adapter.

forms LiST over most of the tasks no matter how many labeled training examples.

**Combination of Different Parameter-Efficient Learning Paradigms in Self-training.** We aim to explore how PEL performs in the self-training procedure. We integrate the PEL paradigm into the teacher or student model to show the performance of the different combinations of PEL. As shown in Table 2, we choose Head-Adapter and Prompt-Adapter. We find the setting that all parameters in both the teacher and student updated gains the best-average performance, indicating the ceiling performance of each paradigm. Yet, it costs about 11 hours which makes the self-training procedure inefficient. In addition, the time influence on whether the teacher model uses PEL is less than the student, because the teacher model only trains once while the student model needs to update for 100 epochs in each self-training iteration. Correspondingly, this motivated us to leverage PEL in the student model to improve the efficiency of self-training, preserving its effectiveness.

| Selection Strategy | Avg. Results |
|---|---|
| None | 76.0 |
| $\alpha = 0$ (w/o. Confidence) | 77.2 |
| $\alpha = 0.2$ | 77.9 |
| $\alpha = 0.4$ | 78.2 |
| $\alpha = 0.6$ | 77.6 |
| $\alpha = 0.8$ | 77.3 |
| $\alpha = 1.0$ (w/o. Certainty) | 76.8 |

Table 4: The average performance (%) of UPET (Prompt-Ptuning) with different selection strategies (varying by $\alpha$). "None" equals Prompt ST which trains the student model on all pseudo-labeled data.

| Methods | SST-2 (acc) | MNLI (acc) | QNLI (f1) | MRPC (acc) | RTE (acc) | CB (acc) | AGNews (acc) | Avg. |
|---|---|---|---|---|---|---|---|---|
| *Prompt-Ptuning* | | | | | | | | |
| Prompt ST | 91.0 | 57.7 | 67.8 | 81.0 | 57.9 | 77.7 | 88.8 | 74.6 |
| UPET | **91.7** | **69.5** | **71.9** | **83.7** | **60.8** | **80.4** | **89.6** | **78.2** |
| w/o. Reliable Example Sampling | 91.3 | 63.0 | 69.8 | 82.2 | 58.3 | 78.3 | 89.2 | 76.0 |
| w/o. certainty | 91.4 | 65.8 | 70.4 | 82.8 | 59.0 | 78.6 | 89.5 | 76.8 |
| w/o. confidence | 91.6 | 66.3 | 71.0 | 83.3 | 59.7 | 78.8 | 89.5 | 77.2 |
| w/o. PHCE loss | 91.3 | 67.2 | 69.3 | 83.0 | 59.9 | 79.7 | 89.3 | 77.1 |
| w/o. Easy-Hard Contrastive Tuning | 91.5 | 65.8 | 68.5 | 82.8 | 58.9 | 79.1 | 89.6 | 76.6 |

Table 5: The 16-shot performance (%) of different variants of UPET with Prompt-Ptuning.

| Methods | #Example | Accuracy |
|---|---|---|
| Variational Pre-training | 200 | 83.9 |
| Reinforcement + Adv. Training | 100 | 81.7 |
| SeqSSL + Self-training | 100 | 78.5 |
| SeqSSL | 100 | 76.2 |
| SeqSSL + Adv. Training | 100 | 76.0 |
| UPET (worst) | 64 | 89.6 |
| UPET (best) | **64** | **91.0** |

Table 6: Performance comparison over AGNews task with non-BERT-based SSL approaches (Li and Ye, 2018; Gururangan et al., 2019; Dai and Le, 2015; Li and Sethy, 2020) (RL: Reinforcement Learning, Adv.: Adversarial, Temp. Ens.: Temporal Ensemble, Layer Part.: Layer Partitioning). UPET (worst) and UPET (best) denote the performance of Prompt-Ptuning and Prompt-Adapter.

**Effectiveness of Reliable Example Sampling.** To validate the effectiveness of the RES, we investigate the effect of the balance factor $\alpha$ in Eq. 5 in terms of the average performance. From Table 4, it is necessary to perform sample selection to obtain more clean data. The results also illustrate that both model confidence and certainty substantially make contribute to the performance. We find the best value is set around 0.2, which means certainty plays an important role in the selection.

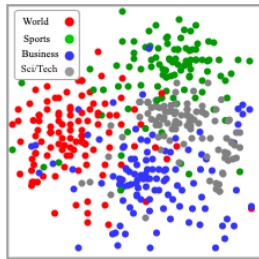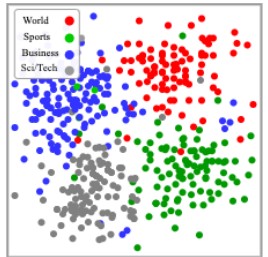

Figure 4: The AGNews's t-SNE visualization of UPET w/o. Easy-Hard Contrastive Tuning (left) and w/ Easy-Hard Contrastive Tuning (right).

**Visualization of the Contrastive Regularization.** To investigate how the proposed Easy-Hard Contrastive Tuning contributes to the final performance, in Figure 4, we use the t-SNE (Van der Maaten and Hinton, 2008) tool and select the AGNews task for validation. Specifically, we randomly sample 1k testing examples to draw the representations in the semantic space. Results demonstrate that the model trained with contrastive regularization can make a clearer boundary between every two classes, corroborating our conclusions that avoiding the overfitting problem and yielding better generalization.

### 4.5 Ablation Study

In this section, we conduct an ablation study to demonstrate the impact of different variants of UPET that remove the designed technique. From Table 5, we thus make the following summarization. 1) We find that the performance of w/o. Reliable Example Sampling (RES) decreases a lot (more than 2%). In addition, we also find that the sampling weight considered by both certainty and confidence can make consistent contributions in RES. These phenomena demonstrate the effectiveness of the de-noising approach considered by both model confidence and certainty. 2) Removing PHCE loss from UPET in 1.1% performance drop in terms of average results, which indicates the importance of PHCE loss in robust student training. 3) Through UPET versus UPET w/o. Easy-Hard Contrastive Tuning, the average performance of the student model is improved by about 1.6%, demonstrating the effectiveness of the contrastive regularization design.

### 4.6 Comparison to Non-BERT Approaches

We end this section with an additional comparison between UPET and non-BERT semi-supervised learning approaches that use a different number

of labeled examples for tuning the teacher model. Table 6 shows that our framework achieves a large performance gain with only 64 labeled examples, especially on UPET (best) with at least 7%.

## 5 Conclusion

In this paper, we introduce a novel uncertainty-aware parameter-efficient self-training framework (UPET) to better improve the effectiveness and efficiency of self-training. In UPET, we use uncertainty estimation to judiciously select reliable pseudo-labeled examples to explicitly alleviate the noisy label problem. To make self-training more efficient, we integrate multiple parameter-efficient paradigms into self-training. To further improve the performance, we also present Easy-Hard Contrastive Tuning to enhance the robustness and reduce the over-fitting problem. In the future, we will extend our framework to other complex tasks, such as sequence labeling, question answering, etc.

## Limitations

Our limitations are shown below:

- We only focus on sequence classification-style NLU tasks. However, we think it can be extended to other tasks easily, such as sequence labeling, question answering, etc.

- Our work focuses on the PLM without Transformer decoders. We think it is possible to extend our method to natural language generation (NLG) tasks. We will leave it as our future work.

## Ethical Considerations

Our contribution in this work is fully methodological, namely uncertainty-aware parameter-efficient self-training (UPET) to improve effectiveness and efficiency based on PLMs. However, transformer-based models may have some negative impacts, such as gender and social bias. Our work would unavoidably suffer from these issues. We suggest that users should carefully address potential risks when the UPET models are deployed online.

## Acknowledgements

This work has been supported by the National Natural Science Foundation of China under Grant No.U1911203, and the National Natural Science Foundation of China under Grant No.62377012.

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

# A Background Knowledge

## A.1 Parameter-Efficient Learning Paradigms

PEL aims to update the partial parameters of the PLM to improve the training efficiency (Mao et al., 2022). We first introduce three classic parameter-efficient methods.

**Ptuning.** Ptuning (Liu et al., 2021b) adds a continuous prompt into the input and uses a prompt encoder to realize parameterization. Specifically, for each input sequence $X$, we have a task-specific prompt template $\mathcal{T}$ as follows:

$$P_1, \cdots, P_I, X, \texttt{It was MASK.}$$

where $P_i$ is a prompt pseudo token (as proposed in Liu et al. (2021b)), $I$ is the total number of pseudo tokens, and MASK is a special token as the placeholder for model output.

**Prefix-tuning.** Prefix-tuning (Li and Liang, 2021) (it can also be viewed as P-tuning V2 (Liu et al., 2021a)) extends the key and value matrix with new continuous vectors in each transformer layer. We also denote the length of the prefix vectors as $I$.

**Adapter.** Adapter-tuning (Houlsby et al., 2019) designs multiple adapter networks into the transformer bloc, which can be viewed as two feed-forward projections. Specifically, the adapters first project the original $d$-dimensional features into a smaller dimension, $m$, apply a nonlinearity, and then project back to $d$ dimensions. The total number of parameters added per layer, including biases, is $2md + d + m$. By setting $m \ll d$, we limit the number of parameters added per task.

We extend these methods into two paradigms, i.e. Head-Tuning and Prompt-Tuning. Head-Tuning aims to stack the prediction layer based on the CLS head, while Prompt-Tuning aims to reuse the pre-training objective of Masked Language Modeling (MLM) and predict by the well-designed template and verbalizer. As shown in Figure 2, we can unify all parameter-efficient methods with both Head-Tuning and Prompt-Tuning.

## A.2 Bayesian neural network (BNN)

We provide a brief introduction to BNN (Mukherjee and Awadallah, 2020). Given a neural model $f^W$ and a training set $\mathcal{D}_l$, the parameters $W$ can be optimized by the posterior distribution $p(W|\mathcal{D}_l)$. In the inference stage, suppose that we aim to generate the label for the unlabeled example $X_i \in \mathcal{D}_u$, we can calculate the probability distribution by:

$$p(y = c|X_i) = \int_W p(y = c|f^W(X_i)p(W|D_u)dW. \quad (12)$$

In other words, BNN averages over all the possible weights instead of directly optimizing for the weights (Mukherjee and Awadallah, 2020). Yet, it is intractable in practice for Eq. 12, so that we can find a surrogate distribution to make the calculation tractable. Specifically, we consider $q_\theta(W)$ to be the dropout distribution (Srivastava et al., 2014) which aims to sample $T$ masked model weights from the current model. Hence, the approximate posterior for each unlabeled example is:

$$p(y = c|X_i) \approx \frac{1}{T} \sum_{t=1}^{T} p(y = c|f^{\widetilde{W}_t}(X_i)), \quad (13)$$

where $\{\widetilde{W}_t\}_{t=1}^{T} \sim q_\theta(W)$ are the masked model weights.

## A.3 Partially Huberised Cross-Entropy

Partially huberised cross-entropy loss (PHCE loss) can be used to alleviate the noisy label problem

| Category | Dataset | #Class | #Train | #Test | Type | Labels (classification tasks) |
|---|---|---|---|---|---|---|
| | SST-2 | 2 | 6,920 | 872 | sentiment | positive, negative |
| | MRPC | 2 | 3,668 | 408 | paraphrase | equivalent, not_equivalent |
| Text | MNLI | 3 | 392,702 | 9,815 | NLI | entailment, neutral, contradiction |
| | QNLI | 2 | 104,743 | 5,463 | NLI | entailment, not_entailment |
| Classification | RTE | 2 | 2,490 | 277 | NLI | entailment, not_entailment |
| | CB | 3 | 250 | 57 | NLI | entailment, neutral, contradiction |
| | AGNews | 4 | 120,000 | 7,600 | topic cls. | world, sports, business, technology |

Table 7: The statistics of multiple languages understanding tasks. Since the original test data is unavailable, we use the development sets as our test sets.

via a simple variant of gradient clipping for the classification loss (e.g. cross-entropy). Given one example $(x, y)$, the PHCE loss $\phi(x, y)$ is denoted as:

$$
\begin{cases}
-\tau p_W(x, y) + \log \tau + 1 & p_W(x, y) \le 1/\tau; \\
-\log p_W(x, y) & p_W(x, y) > 1/\tau;
\end{cases}
\tag{14}
$$

where $\tau > 1$ is the hyper-parameter. Thus, the model learned by Eq. 14 can be more robust to the noisy labeled tokens than the common cross-entropy.

## B Self-training Procedure

We show the whole training procedure in Algorithm 1. Specifically, we first use the original PLM $f^{W_0}$ to initialize a teacher model $f^W_{tea}$ (Algorithm 1, Line 1), and then fine-tune the teacher model over few-shot labeled data $\mathcal{D}_l$ (Algorithm 1, Line 2). During the iteration process, we sample a subset unlabeled set $\mathcal{D}'_u$ from $\mathcal{D}_u$, and obtain model confidence and certainty for each unlabeled example $\widetilde{X}$ (Algorithm 1, Line 4). Based on these factors, we can calculate the sampling weight for each unlabeled example and sample some reliable examples to form an easy set $\mathcal{D}_r$, and the rest is formed as a hard set $\mathcal{D}_h$ (Algorithm 1, Line 7-9). During the student learning, we use the original PLM $f^{W_0}$ to initialize a student model $f^{w^*}_{stu}$, and use PHCE loss and Easy-Hard Contrastive Tuning to train the parameter-efficient student model over the pseudo-labeled examples (Algorithm 1, Line 6, 10-12). At last, we can copy the parameter of the student model to the teacher and repeat until convergence.

## C Details of NLU task

We list the statistics of each task in Table 7.

| Teacher Hyper-parameter | Value |
|---|---|
| Batch Size | {4, 8} |
| Seed | {12, 21, 42, 87, 100} |
| # Examples per Class | {16, 32, 64} |
| $\alpha$ | {0.1, 0.3, 0.5, 0.7, 0.9} |
| $\gamma$ | {0.001, 0.01, 0.05, 0.1, 0.5, 1.0} |
| Prefix Length $I$ | {4, 8, 16, 32, 64, 128} |
| Adapter Small Dim $m$ | {8, 16, 32, 64, 128, 256} |

Table 8: The searching scope for each hyper-parameter.

## D Searching Scope of Grid Search

We use grid search to select the best hyper-parameters for each task, the searching score is shown in Table 8.

---

**Algorithm 1** Self-training Procedure of UPET

**Require:** Neural model $f^{W_0}$, labeled data $\mathcal{D}_l$, unlabeled data $\mathcal{D}_u$.
1: Initialize a teacher model $f^W_{tea} = f^{W_0}$;
2: Fine-tune the teacher model $f^W_{tea}$ over the labeled data $\mathcal{D}_l$ (All parameters will be updated);
3: **while** not converged **do**
4:     Sample an unlabeled data subset $D'_u \subset D_u$;
5:     Pseudo annotate each unlabeled example $\widetilde{X}_i \in \mathcal{D}'_u$ by $f^W_{tea}$ in Eq. 1 to obtain the hard label $\widetilde{Y}_i$;
6:     Initialize a student model $f^{W^*}_{stu} = f^{W_0}$;
7:     Obtain the certainty score $s^{ct}_i$ for $\widetilde{X}_i$;
8:     Obtain the confidence score $s^{cf}_i$ for $\widetilde{X}_i$;
9:     Sample reliable examples to form a subset $\mathcal{D}_r$ by the sampling weight in Eq. 5. The examples not sampled can be used to form $\mathcal{D}_h$.
10:     Calculate the PHCE loss $l(\mathcal{D}_r, f^{W^*}_{stu})$ in Eq. 9;
11:     Calculate the regularization loss $R(f^{W^*}_{stu})$ in Eq. 10;
12:     Training $f^{W^*}_{stu}$ via parameter-efficient learning by reduce $\mathcal{L}(\mathcal{D}_r, f^{W^*}_{stu})$ in Eq. 11;
13:     Update the teacher model $f^W_{tea} = f^{W^*}_{stu}$;
14: **end while**
15: **return** The teacher model $f^W_{tea}$.