# OpenReview forum: "Uncertainty-aware Parameter-Efficient Self-training for Semi-supervised Language Understanding"
_EMNLP/2023/Conference — EMNLP 2023 Findings_

### Official Review · Reviewer_rnmD · 2023-07-20

**Paper Topic And Main Contributions:** 1. The development of UPET, an uncert…
**Soundness:** 2

**Excitement:**

3: Ambivalent: It has merits (e.g., it reports state-of-the-art results, the idea is nice), but there are key weaknesses (e.g., it describes incremental work), and it can significantly benefit from another round of revision. However, I won't object to accepting it if my co-reviewers champion it.

**Questions For The Authors:**

 I will reconsider my score in the rebuttal.

**Reasons To Accept:**

1. The paper addresses a significant problem in the field of NLP, i.e., the dependence of large pre-trained language models on extensive labeled data, which is often not available in low-resource scenarios.
2. The proposed UPET model innovatively incorporates uncertainty estimation and an efficient parameter-updating method into the self-training loop, leading to better handling of noise and computational efficiency.
3. The paper provides extensive experimental results, showcasing the effectiveness of UPET in various language understanding tasks and in different resource settings. It demonstrates that UPET can outperform baseline methods, indicating its practical utility.
4. The methodology and experimental setup sections of the paper are detailed and clear, making it possible for other researchers to reproduce the results.


**Reasons To Reject:**

1. While the paper does an excellent job of demonstrating the effectiveness of UPET in low-resource scenarios, it could benefit from more discussion and testing in high-resource scenarios. Understanding how UPET performs when abundant labeled data is available would provide a fuller picture of its versatility and performance.
2. The paper could further discuss and explore the choice of parameters that are updated during the efficient training method. A more detailed analysis on this aspect could provide insights into why certain parameters are more beneficial to update and how it affects the overall performance.
3. The paper could also benefit from comparisons with other semi-supervised learning methods or other parameter-efficient training methods. This would help to position UPET within the broader field of semi-supervised learning for NLP.
4. While the paper does present results on several tasks, including more diverse tasks or languages could further demonstrate the robustness and generalizability of UPET.

**Reproducibility:**

3: Could reproduce the results with some difficulty. The settings of parameters are underspecified or subjectively determined; the training/evaluation data are not widely available.

**Reviewer Confidence:**

4: Quite sure. I tried to check the important points carefully. It's unlikely, though conceivable, that I missed something that should affect my ratings.

---

> ### Author Rebuttal · Authors · 2023-08-28
>
> We truly thank you for your constructive criticism and valuable suggestions as to how to make this paper stronger.
>
> > R1: While the paper does an excellent job of demonstrating the effectiveness of UPET in low-resource scenarios, it could benefit from more discussion and testing in high-resource scenarios. Understanding how UPET performs when abundant labeled data is available would provide a fuller picture of its versatility and performance.
>
> Thank you for your suggestions. In this work, we mainly focus on low-resource scenarios, which are more challenging than high-resource scenarios. If the labeled data is abundant, we think the model can better capture semantics and there is no need to perform semi-supervised learning. To this end, we will conduct additional experiments using more labeled data / more shots and analyzing UPET's performance under identical settings.
>
> > R2: The paper could further discuss and explore the choice of parameters that are updated during the efficient training method. A more detailed analysis on this aspect could provide insights into why certain parameters are more beneficial to update and how it affects the overall performance.
>
> Thank you very much for your suggestion regarding the choice of parameters that are updated. Due to space constraints, we primarily focus our analysis on self-training and label efficiency to demonstrate the rationale behind our main framework. Following your advice, we will add a new set of experiments (along with statistics) to the appendices (or the additional one page in the camera-ready version) in our next revision, analyzing why certain parameters are more beneficial to update and how they will impact overall performance.
>
> > R3: The paper could also benefit from comparisons with other semi-supervised learning methods or other parameter-efficient training methods. This would help to position UPET within the broader field of semi-supervised learning for NLP.
>
> We have conducted some experiments as the reviewer mentioned. For the comparison with other semi-supervised learning methods, the results in Table 6 show that our method can achieve the best performance. For the different PEL methods, we have conducted the comparison on six different PEL settings in Table 1. The results demonstrate that P-tuning with the Prompt paradigm can outperform other PEL methods.
>
> > R4: While the paper does present results on several tasks, including more diverse tasks or languages could further demonstrate the robustness and generalizability of UPET.
>
> Thank you for your suggestion. We will conduct experiments over other NLP tasks, such as question answering, text generation, and translation. Specifically, in the final version, we will incorporate tasks like SQuAD as further evidence of UPET's effectiveness.
>
> We sincerely appreciate your insightful and valuable comments/criticisms, and we are very grateful for your willingness to reconsider the score in the rebuttal. Thank you for your suggestions that can make this paper stronger.

---

### Official Review · Reviewer_exqi · 2023-08-04

**Soundness:** 4

**Excitement:**

4: Strong: This paper deepens the understanding of some phenomenon or lowers the barriers to an existing research direction.

**Missing References:**

Following two papers might be worth citing:

- Active bias paper – https://arxiv.org/abs/1704.07433 – Haw-Shiuan Chang, Erik Learned-Miller, Andrew McCallum, Active Bias: Training More Accurate Neural Networks by Emphasizing High Variance Samples, NeurIPS 2017.
- Hard example mining – A. Shrivastava, A. Gupta, and R. Girshick. Training region-based object detectors with online hard example mining. In CVPR, 2016

**Paper Topic And Main Contributions:**

This paper proposes a new self-training method that uses an uncertainty-aware machine learning model, specifically Bayesian neural networks (BNNs). The method first exploits the model's confidence and uncertainty to select reliable samples from unlabeled data. These samples are then used to perform PEFT-style fine-tuning on the student network. The paper also proposes an easy-hard contrastive tuning method to further improve the performance of the student network. Towards the end, this work presents strong empirical results validating the efficacy of the proposed methods.

**Questions For The Authors:**

- Is PEL the same as PEFT? PEFT is widely accepted in the community IMO. So, you may want to stick with the PEFT term.
- How are proposed ERT methods different from PEFT methods? Are they not the same with the only difference that now your dataset is obtained in SSL setup instead of a manually labeled dataset?
- Did you try to look at the papers on hard example mining for contrastive learning? How does your method compare to other approaches, e.g., the active bias approach or hard mining example approach from the vision domain?
- What does improvement stops after a few iterations? Is it something observed in prior works on self-training as well?

**Reasons To Accept:**

- The paper addresses a very important SSL problem of selecting samples using uncertainty derived from BNNs. While BNNs are well known, this paper presents experiments and results how they can be used for this particular problem in the self-training setup.
- Empirical results are impressive, e.g., Table 1 shows substantive improvements over other baselines.

**Reasons To Reject:**

- The methodological innovations behind PEL / ERT steps seem similar to the PEFT setup, and not clear if it adds any new too or insights for NLP practitioners/researchers.
- The method has many components which makes it harder to gauge the crucial part and relative importance of all the components, especially when practitioners need to apply these methods to novel scenarios, i.e., scenarios not considered in the paper.
- Easy-hard contrastive learning is presented without contextualizing prior work in this area. It’s unclear how and why this method is better than prior works. Empirical evidence to support this part of the proposed work is also not satisfactory as t-SNE plots can be easily misleading.

**Reproducibility:**

2: Would be hard pressed to reproduce the results. The contribution depends on data that are simply not available outside the author's institution or consortium; not enough details are provided.

**Reviewer Confidence:**

4: Quite sure. I tried to check the important points carefully. It's unlikely, though conceivable, that I missed something that should affect my ratings.

**Typos Grammar Style And Presentation Improvements:**

- Line 189: p is the probability distribution but the domain of the distribution is not mentioned clearly. ‘c’ variable is not defined as well. May be you can omit ‘c’ altogether.
- Line 195 and 196: cite in text without brackets.
- Notations are confusing in the methodological section, e.g., sometimes W has tilde over it in line 217 and other times it is missing.
- Writing for the BNN section is unclear and requires substantive gazing efforts. Also, it seems possible to cut few details that are not required to sub-select samples.

---

> ### Author Rebuttal · Authors · 2023-08-28
>
> Thanks for carefully reviewing our paper, we hope our responses can address your concerns.
>
> > Q1: Is PEL the same as PEFT? PEFT is widely accepted in the community IMO. So, you may want to stick with the PEFT term.
>
> Thanks for your query about the terms. PEL is parameter-efficient learning which is the same as PEFT (parameter-efficient fine-tuning). Our proposed module PET belongs to PEL, however, it still has some differences with PEFT: 1) We propose two prediction paradigms, i.e.,  head-tuning (add a new MLP head for cls) and prompt-tuning (add template and verbalizer to reuse MLM head), 2) Our PET can integrate some PEL methods (e.g., Adapters, LoRA from PEFT toolkit) with above tow difference paradigms. For example, we can leverage templates and verbalizers to perform prompting with only tuning adapter modules.
>
> > Q2: How are proposed ERT methods different from PEFT methods? Are they not the same with the only difference that now your dataset is obtained in SSL setup instead of a manually labeled dataset?
>
> Thanks for this question. Our proposed ERT module leverages PEFT to make the training processing more efficient. The main differences are: 1) ERT consists of two main stages, such as parameter-efficient tuning on two prediction paradigms (i.g., Head-tuning and Prompt-tuning); 2) ERT also improves model robustness when training on pseudo-labeled data by using contrastive learning.
>
> > Q3: Did you try to look at the papers on hard example mining for contrastive learning? How does your method compare to other approaches, e.g., the active bias approach or hard mining example approach from the vision domain?
>
> We do not compare other hard example approaches because that the hard sample weight derived from uncertainty estimation can make the framework end-to-end. We do not focus only on hard examples of mining in this work. However, we will add these methods as our new baselines and perform external experiments in the revised version.
>
> > Q4: What does improvement stop after a few iterations? Is it something observed in prior works on self-training as well?
>
> Thanks for your astute observations. The improvement stops when training after about 4 iterations, which suggests that our framework can converge quickly. As shown in Figure 3, especially, when training after one iteration, more reliable unlabeled examples may selected for training the student model. When training more than about 4 iterations, there are fewer unlabeled data that may be useful for improvement. Sometimes, when the iteration number is larger, the model may overfit and make more wrong predictions.
>
> > About Missing References
>
> We sincerely thank you for pointing out these two important papers that are worth citing. We will incorporate them into the related works section in the final version or directly discuss their relevance.
>
> > About Typos Grammar Style And Presentation Improvements:
>
> We sincerely appreciate your attention to detail in pointing out the typos in the notations, as well as alerting us to the citation format errors in Lines 195 and 196. We will make the necessary corrections in the final version. Additionally, we aim to further refine the notations to make the paper more comprehensible. Regarding your feedback on the BNN section, we wholeheartedly agree and will revise this section for clarity.
>
> > Responses to Reasons To Reject
>
> Moreover, we have carefully read your comments in the "reasons to reject." For issues regarding PEL / ERT steps, please refer to our response in Q1. We also appreciate your attention to tasks beyond natural language understanding. We concur and plan to expand our experiments to encompass other NLP tasks such as question answering, text generation, and translation. Specifically, in the final version, we aim to include more tasks like SQuAD as further evidence of UPET's effectiveness.
>
> Regarding your concerns about Easy-hard contrastive learning, under the context of self-training, it is an original idea from us. The setting for t-SNE plots is designed for readers to intuitively grasp our approach. Moreover, we fully acknowledge and support your recommendation on contextualizing prior work. In the upcoming version updates, we will add additional previous works to better position our study within the existing literature.
>
> Thanks again for your careful reviewing and insights!

---

### Official Review · Reviewer_a1vW · 2023-08-05

**Soundness:** 4

**Excitement:**

4: Strong: This paper deepens the understanding of some phenomenon or lowers the barriers to an existing research direction.

**Paper Topic And Main Contributions:**

The authors present an adaptation of the teacher-student paradigm to allow training on massive amounts of unlabelled data in a semi-supervised fashion. This work specifically focuses on natural language understanding (NLU) but they posit their approach could be used for a number of other tasks. The framework presented in this work uses Monte Carlo sampling and Bayesian neural networks in the teacher network to sample high-confidence examples to use in training a student network. Varying the difficulty of examples presented to the student network acts as a regularizer effect to ensure that the student does not overfit on shorter, easier training sequences.

**Reasons To Accept:**

The authors walk through their approaches with clear explanations of the various design choices and mathematical motivations. Experiments are performed against a series of strong baselines and a thorough combination of their approaches to demonstrate gains in F1 scores on a number of natural language understanding test sets and take care to run these experiments a number of times to report statistically significant results.

**Reasons To Reject:**

The authors state in the limitation section that they focused solely on pretrained language models that do not use transformer-based decoders. It would be nice to see results from this side of the 'transformer family tree', but not a show-stopper given the thorough experiments described in this work.

**Reproducibility:**

4: Could mostly reproduce the results, but there may be some variation because of sample variance or minor variations in their interpretation of the protocol or method.

**Reviewer Confidence:**

3: Pretty sure, but there's a chance I missed something. Although I have a good feel for this area in general, I did not carefully check the paper's details, e.g., the math, experimental design, or novelty.

**Typos Grammar Style And Presentation Improvements:**

Tables 1 and 2 have a column for '# Tuable Params' - is this meant to be Tunable Params?

---

> ### Author Rebuttal · Authors · 2023-08-28
>
> We are very grateful for receiving your review, and here are our responses:
>
> > The authors state in the limitation section that they focused solely on pre-trained language models that do not use transformer-based decoders. It would be nice to see results from this side of the 'transformer family tree', but not a show-stopper given the thorough experiments described in this work.
>
> Thank you for reviewing our limitations sections. The pseudo-labeling framework with uncertainty estimation needs to obtain the logits (probability distribution) for T times. It will take too much time when apply for generative LLM. However, if we can obtain the logits for each generated token, the proposed UPET can be easily adapted to generative LLM. We will follow your suggestions and are planning to extend the UPET to more backbones in the future, such as conducting additional experiments using models based on the T5 architecture.
>
> > Typos Grammar Style And Presentation Improvements
>
> We deeply appreciate your meticulous review of our paper and for identifying this issue. Indeed, it's a typo, and we have made the corrections. Thank you again for helping improve our paper.

---

### Meta-Review · Area_Chair_6nNS · 2023-09-16

**Recommendation:** 3

**Metareview:**

This work presents develops an uncertainty-aware and parameter-efficient self-training framework called UPET, which involves using uncertainty estimates to select reliable samples from unlabeled data and introduces an easy-hard contrastive tuning method. The experiments are conducted against a series of baselines and provide statistically significant results that demonstrate gains in F1 scores on a number of natural language understanding test sets. In addition to the positive comments, the reviewers also present some concerns: Reviewer 1 notes that the authors focus solely on pretrained language models that do not use transformer-based decoders,  Reviewer 2 notes that the methodological innovations behind the proposed methods may not add significant new insights or techniques for NLP practitioners and researchers, and Reviewer 3 suggests that the paper could benefit from more discussion and testing in high-resource scenarios, and a more detailed analysis of the choice of parameters that are updated during the efficient training method.

---

### Decision · Program_Chairs · 2023-10-07

**Decision:**

Accept-Findings

**Comment:**

This work presents develops an uncertainty-aware and parameter-efficient self-training framework called UPET, which involves using uncertainty estimates to select reliable samples from unlabeled data and introduces an easy-hard contrastive tuning method. The experiments are conducted against a series of baselines and provide statistically significant results that demonstrate gains in F1 scores on a number of natural language understanding test sets. In addition to the positive comments, the reviewers also present some concerns: Reviewer 1 notes that the authors focus solely on pretrained language models that do not use transformer-based decoders,  Reviewer 2 notes that the methodological innovations behind the proposed methods may not add significant new insights or techniques for NLP practitioners and researchers, and Reviewer 3 suggests that the paper could benefit from more discussion and testing in high-resource scenarios, and a more detailed analysis of the choice of parameters that are updated during the efficient training method.